# A Structural Approach to Some Contradictions in Worldwide Swine Production and Health Research

Juan Felipe Núñez-Espinoza [1,*], Francisco Ernesto Martínez-Castañeda [2], Fernando Ávila-Pérez [3] and María Camila Rendón-Rendón [2]

1  Desarrollo Rural, Colegio de Postgraduados, Campus Montecillos, km 36.5, Carretera Mexico-Texcoco, Texcoco C.P. 56230, Mexico

2  Instituto de Ciencias Agropecuarias y Rurales (ICAR), Universidad Autónoma del Estado de Mexico, Toluca C.P. 50295, Mexico; femartinezc@uaemex.mx (F.E.M.-C.); mcrendon@gmail.com (M.C.R.-R.)

3  Facultad de Medicina Veterinaria y Zootecnia, Universidad Autonoma de Estado de Mexico, Campus El Cerrillo Piedras Blancas, Toluca C.P. 50200, Mexico; favilap095@gmail.com

*  Correspondence: nunezej@colpos.mx

**Abstract:** Several biosafety gaps in agri-food sectors have become evident in recent years. Many of them are related to the global livestock systems and the organizational models involved in their management and organization. For example, producing pigs requires a global system of massive confinement and specific technological innovations related to animal production and health that involve broad technical and scientific structures, which are required to generate specific knowledge for successful management. This suggests the need for an underlying socially agglomerated technological ecosystem relevant for these issues. So, we propose the analysis of a specialized scientific social structure in terms of the knowledge and technologies required for pig production and health. The objective of this work is to characterize structural patterns in the research of the swine health sector worldwide. We use a mixed methodological approach, based on a social network approach, and obtained scientific information from 4868 specialized research works on health and pig production generated between 2010 to 2018, from 47 countries. It was possible to analyze swine research dynamics, such as convergence and influence, at country and regional levels, and identify differentiated behaviors and high centralization in scientific communities that have a worldwide impact in terms of achievements but also result in significant omissions.

**Keywords:** biosafety gaps; swine global production; social network analysis

## 1. Introduction

Pork is the second most consumed and commercialized meat worldwide [1]. In order to meet this demand, thousands of hectares and a variety of processes and international protocols are used to produce the pork consumed in such a wide range of markets [2]. Therefore, pig farming has experienced exponential commercial growth in terms of the numbers of animals, the size of production units, their yields and degree of specialization [3]. At the same time this growth has generated intense competition for local, regional and worldwide markets [4,5], as well as more concentrated production.

This is exemplified by the large corporations that "dominate" not only regional markets but also the world market. The main pork producers and processors in the United States are Smithfield Foods, which had an inventory of 1,241,000 breeding sows in 2019, and Triumph Foods, which in that same year had 487,200 breeding sows. Among the leading Asian companies, in terms of pork production, the Wens Foodstuff Group had a total of 1,200,000 breeding sows that produced 22.3 million pigs for human consumption in 2018, and CP Foods, with 800,000 breeding sows produced 2.8 million animals. The South American firm BRF, which is a fusion of Brazil's leading agri-food companies Sadia and

Perdigão, has 400,000 breeding sows. In the European Union, Cooperl (251,000 breeding sows) and Vall Companys (213,000 breeding sows) had a combined output of 10 million pigs [6].

The total value of worldwide production of pork and edible offal (without considering fat) reached the sum of USD 255,682,742,000.00 in 2019 [1]; China contributed with 45.89%, followed by the United States with 9.20% and Germany with 3.57%. Likewise, China held first place in terms of the volume of output, generating 39.32% of the world total, followed by the US with 11.58% and Germany with 4.53% [1].

Commercial pig production has intensified significantly, producing a greater number of pigs in smaller spaces and with an increase in the yield of animal origin products. This means large-scale production systems with a high degree of uniformity in terms of genetics, feeding and infrastructure for animal handling. Therefore, despite the increases in productivity, profitability and improvement in the health status of stabled herds, and intensive production, the mere size of the world pig farming system entails significant contradictions in at least two aspects: in managing the waste it generates, and as the place of origin for various zoonotic-based scenarios in the world.

In relation to the latter, a large number of infectious diseases found in humans have a zoonotic origin and are directly related to modern livestock production technologies. The invention of mass confinement, as a strategy to improve exploitable animal species, has increased productivity and livestock control as never before, but it has also given rise to the presence of large populations of immunosuppressed animals, generating optimal conditions and spaces for more rapid cultivation and dissemination of pathogens. If we add to this the expansion of densely populated urban human nuclei, with high rates of production and consumption of products of animal origin, we generate the perfect storm for dispersing biological agents of all kinds [7]: Many of the diseases that human populations currently suffer from are derived, directly or indirectly, from the food system [8].

In the case of the waste produced by this enormous pig production, there has been a significant increase in the leachate dumping and generation of greenhouse gases, such as $CO_2$, $CH_4$ and $N_2O$ (Nitrous oxide), both have caused irreparable damage to the biotope and the ozone layer [9–11]. These problems are reproduced even more acutely in the more developed economies.

In the same context, it is also important to point out the other paradox of pig farming agribusiness: the relationship with the generation and export of zoonotic systems; in other words, the occupational diseases resulting from professional practice whereby a person may be more exposed to certain diseases than the general population [12]. For example, the G4 strain of swine flu has been transmitted to human populations making them ill (having as the first source of contagion the personnel in charge of swine production) and achieving the potential to become a "pandemic virus" [13]. Furthermore, the spectrum broadens to include other possible infections such as Erysipelas, Leptospirosis, etc.

The other problem is the frequency and presence of diseases that affect pigs, which continue to be one of the biggest challenges and paradoxes for this sector, mainly because in recent decades there has been an increase in the number of biosecurity gaps in the management of already established, massive herds of pigs, as well as the global incursion of virus strains in new geographical territories, including Asia, Eastern Europe, Africa, South America, etc. [14,15]. The growth of herds and the standardization of livestock production systems are at the root of these problems.

The African swine fever (ASF), which has affected Asian countries since 2018 (China, the Philippines and Vietnam), has caused a considerable decline in pig production in those regions. For example, in China in 2019, output totaled 42.6 million tons; that year showed the most drastic decrease in production on record, a decline of 21.3%. It was caused by the impact of this disease on their herds and the more rigorous environmental measures that regulate the operation of pig-producing farms. That country's pig inventory has been declining significantly, thereby affecting production levels. Currently ASF continues to spread, although at a slower pace, which has led small and medium producers to remain cautious in deciding to repopulate and/or increase their respective herds [16].

Another pathology that currently appears to accompany pig production is the porcine reproductive and respiratory syndrome (PRRS) that affects breeding herds and growing pigs; it is measured by a decline in reproductive health, an increase in deaths and reductions in the rate and efficiency of growth [17]. The total nationwide cost of productivity losses in breeding herds and growing pigs in the US, due to PRRS, has been estimated at USD 664 million (US) per year, which is a sizeable increase from the USD 560 million estimated in 2005. The 2011 study differed most significantly from the 2005 study in terms of the distribution of the losses between breeding herds and growing pigs. Losses in breeding herds accounted for 12% of the total cost of PRRS in the 2005 study, compared to 45% in the current analysis [18].

Furthermore, classical swine fever (CSF) is one of the most relevant viral epizootic diseases found in pigs. Given its severe economic impact, it is mandatory that detection of classical swine fever be notified to the World Organization for Animal Health (OIE). Despite the fact that it has been controlled in many parts of the world, the costs derived from the last two outbreaks of classical swine fever in Spain (1997 and 2001) were approximately 108 million euros [19].

Respiratory diseases in swine have a considerable economic and productive impact on the industry, lowering feed efficiency and growth rates, and causing higher morbidity and mortality rates, increased costs of drugs and lower carcass quality. Pigs affected by enzootic pneumonia, for example, tend to eat less than is necessary for their maintenance and development. The lack of nutrients leads to the inability to reach their maximum genetic potential for muscle synthesis and fat accumulation. These metabolic alterations reduce live weight and decrease carcass quality, provoking losses of up to USD 6.55 per pig at the time of slaughter [20].

The aforementioned aspects bring together three central issues for this sector: production, health and social structure. Therefore, we can safely presume close interaction between two highly visible sectors: meat production and pharmaceuticals (mainly animal health-veterinary services). This structure of connections between the health sector and the meat sector propitiated a turning point for the veterinary medicine industry. This industry has had sustained growth in recent decades [21], promoting the development and diversity of medicines at a regional level [22] as well as reducing costs [23] and increasing volumes applied [24].

Moreover, the pharmaceutical industry has enabled itself to massively sell drugs for all livestock species and one of the consequences of this is that meat products and derivatives for human consumption have become contaminated with antibiotics and other antimicrobials [25]. In a similar fashion, the broad range of by-products and/or derivatives of pork production, used by the food industry itself, as well as other industries, (manufacture of processed foods, food for animal consumption and diets, pharmaceutical, cosmetic and chemical industries, among others) [26] led to the establishment of compromised structural correlations, for example, related to the culture of consuming processed meats (bacon, sausages, hot dogs, salami, ham, pepperoni, various types of cold cuts, etc.) and/or red meats with an increase in the rates of colorectal cancer, pancreatic and prostate cancer in human populations [27].

In 2020 when the WHO declared that COVID-19 was a pandemic, this organization indicated, in the midst of the outbreak, that it was not only a health crisis but, in fact, a structural one [28], affecting all the social and economic sectors of society, given the high resonance it was having throughout the entire global social structure. Thus, the emergence of the pandemic led to efforts to determine the outbreak's origins. These efforts pointed to two possible specific sites of social agglomeration: a market that sells the meat of wild animals, the Wuhan City 'wet' market (Huanan Wholesale Seafood Market) [29], and a specialized research center, (Wuhan Institute of Virology) [30]. While the possible origins of the pandemic are still in question, the trade–research relationship has high probabilities for generating problematic scenarios such as the one currently underway [31–33]. This suggests that there might be a correlation between biosecurity gaps and the trade-research binomial [34–37].

The invention of mass confinement, as a strategy to improved and increased meat production, incorporated specific and unprecedented technological and organizational developments, leading to an overall increase in the productivity of these systems. However, it also generated wide gaps in biosecurity in the control and management of massive herds, which have posed systemic health dilemmas for the human population. In this direction, the dispersion of zoonotic systems in the world can be associated with the massive confinement systems prevalent in livestock production today, due to two general characteristics: their scope and regional inter-connections, and the uniformity of their productive structures This presents us with the possibility of increasingly recurring biosafety gaps, which is why it is suggested that they are not random and/or esoteric events but rather part of a specific order of social systems.

In relation to the pig sector, it should be noted that most of its technological knowledge is generated in delimited relationships between public and private institutions, and involved regional, national and supranational swine research systems. This leads us to consider various confluent, interrelated, inter-overlapping and overlapping social structures: the economic structures of the centers for production, transformation and collection [38]; industrial and agroindustry structures [39]; sanitary structures [40]; legislative structures (laws, regulations); and research and education structures. The latter is a labile structure that horizontally runs through the adjacent intricacies of the other social structures, appearing on a recurring basis, given its responsibility for the training and management of specialized human resources (technicians, consultants, researchers, officials, etc.) and involvement in the generation of metrics and/or measurement parameters and/or analysis of the dynamics of different productive sectors. Therefore, they are a central part of the social system analyzed. According to Núñez [41], these types of research and educational structures are involved in community dynamics (within societies, communities, towns, countries, etc.) to process, adapt and/or mold public opinion and preferences (the ideological system) in favor of one technological path or another.

Based on this, we suggest that the biosecurity deficits outlined above, in a system such as pig farming, have a point of origin in the decomposition that exists at the institutional level (public and/or private). In this regard, the FAO [42] indicates that the construction of destructive events (such as a pandemic) is linked to: (1) institutional deficiencies for guaranteeing safe and equitable access to consumption inputs (for example, information); (2) insufficient (and deficient) infrastructure needed to supply the market with quality products; and (3) mismatches between supply and demand. In this context, the presence of deficient institutional systems generates conditions for socio-structural vulnerability [43], so that the breakdown of the systems for analyzing and/or evaluating biosecurity in pig production systems could indicate, in turn, dynamics of institutional decomposition (public and/or private) [44].

In this sense, it is important to point out that the pig sector, given its deep involvement in a wide range of social, economic and productive sectors, etc. worldwide, has a distinctive social multiplier factor. For example, in countries such as Mexico, this means that for every million pesos invested in the pig production sector 1.24 million pesos are generated in other sectors providing inputs for this sector, and 160 million pesos in output are produced [39]. In this way, pig production is articulated with a variety of productive chains, and various actors and/or areas (commerce, medicine, research, teaching, etc.), which, as social conglomerates, in their productive interactions, produce structural patterns that are possible to register, analyze and measure. Therefore, given the range of problems the sector must confront, there are many areas to be researched and opportunities for processing the information circulating within the social structures of pig production.

The foregoing opens up significant space for research on these topics, mainly because we believe that, given the problems we have pointed out, there are underlying research and educational structures of adjacent actors and entities that have generated a social ecosystem that supports, by default, an intertwined structure of production and animal health systems. Furthermore, such underlying structure can be observed—and measured—

from the dynamics of scientific agglomeration and collaboration which offers analytical elements to understand social behavior from both technicians and academic sector which are part of those overlapping structures that confluent in the strategy for improved and increased swine production.

Henceforth, it is legitimate to try to characterize the dynamics of such a social structure by posing the following questions: What are the organizational patterns that prevail in the social structure responsible for research on health issues within the pig production sector? What are the social dynamics within the regional and national structures responsible for pig research? Based on these questions, the main objective of this work is to recognize and characterize socio-structural elements in the field of research on health-related issues in pig production at the global, regional and national levels.

We need to clarify that, given the limited scope of this paper, we are not aiming to analyze the global research system. It is not possible for us to consider "all" of the already existing research nor that which is currently underway, except with a very delimited, synthesized and specified exercise. However, this does not imply that doing so is not achievable. Therefore, we intend to raise issues and questions that need to be explored as part of such a process. That is why we are proposing the methodological approach we present here.

## 2. Materials and Methods

Every agricultural sector or process is generated through the interaction, organization and overlapping of various productive chains, each of which contains innumerable social actors (producers, laborers, companies, technicians, organizations, specialists, etc.). Thus, they can be conceptualized as systems and/or structures for communicating and transmitting all kinds of information (linguistic, biological, political, etc.). Productive chains are present in multiple communities that constitute a system for exchanging information in one productive direction or another. Whatever affects a sector is immediately transmitted to all of the other sectors with which it is connected. This is indicative of underlying communication structures that are not necessarily obvious. It is possible to visualize one of these structures through collaboration via scientific activities between specialized human resources (technicians, consultants, researchers, etc.), for instance, research projects, books, publications in peer-reviewed journals or in conference papers, among others. Most of these products contain a specific particularity: they were written by more than just one author, so this partnership allows the understanding part of the complexity of one of the most intelligible and well-known forms of collaboration structures because the co-authorship defines a coherent and reasonable social scientific formula. Confidence is essential at the moment of writing together a scientific paper [45]. Additionally, the co-author networks have become much more complex and wider than before, so it is possible to suggest organizational patterns by scientific areas within such networks; therefore, it is legible to suggest communities and clustering dynamics of the knowledge by scientific field [46]. In this context, the majority of the network analysis about co-authorship has been related to the writing (and citation) of books and publications in peer-reviewed journals [47–49]; however, there is a scientific collaboration form that has not been commented enough or at least as much as the latest: the co-authorship in conference papers. The participation in any conference, international congress or local forums, etc., begins with some rational, strategic and specific ideas: the field of knowledge to participate, the kind of ideas and research results to share and in what group it is possible to participate [50], and all of this, because each conference could provide some opportunities to start building a scientific network, develops science communication skills (advocating for your research field) and permits the access to benefits such as an increase in the visibility of your research, project collaboration, possible access to research funds, professional transitions and learn valuable information from others researchers working with common research objectives. Most importantly, this would foster friendships and confidence with others similar researchers [51,52].

Furthermore, each congress or conference is not just an academic event, but it is a social, economic and even a political meeting as well; it even could be seen as an interstice dialogue between communities to facilitate political distension among people and institutions. We suggest that there are much more possibilities to fertilize the knowledge because of the temporal quality of the communities at conferences especially because the weak ties system that prevalence in there [53]. In the conferences, the scientific communities explore new research resources and social bonds and given the size of the academic agglomerations it is possible to identify social and massive movements as well as the evolution of research groups, particularly with the co-authorship formula. In this direction, it would be convenient to distinguish this kind of social particularity in the collaboration network analysis.

The subject matter presented was analyzed from a mixed approach using two basic tools:

1. The set of swine scientific research documents published in the proceedings of the International Pig Veterinary Society (IPVS), between 2010 and 2018 [54–58], and which are public access documents too (http://www.theipvs.com/links/ (accessed on 5 May 2020). IPVS represents a historical model for the integration of veterinary and pig research. It was established in 1969 with the aim " . . . to share knowledge related to pig health and production and to foster potential cooperation among pig veterinary societies, scientists, swine veterinarians and pork producers . . . " [59]. The IPVS's main objectives include: (1) the exchange and analysis of knowledge related to *pig health and production*; (2) and the formation of *Pig Veterinary Societies* in all pig-producing countries and promotion of cooperation between such societies [60]. These antecedents constitute the basis for the association's convening capacity, which can be observed in the growing scientific community that has participated in the 5 congresses that produced the proceedings selected for this analysis (Figure 1).

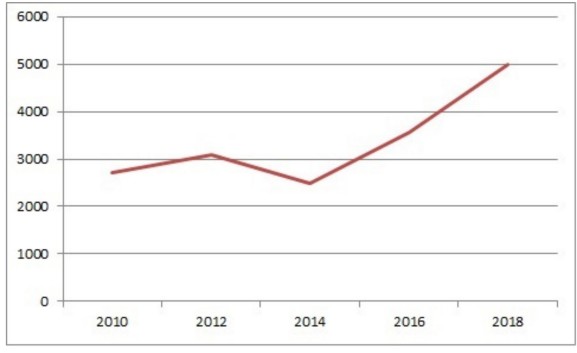

**Figure 1.** Tendency of growth in the participation in IPVS congresses (2010–2018). Source: [54–58].

It should be pointed out that while this is not the only forum that provides a space for gathering information, analysis and data on swine research, its history as an organization allows us to infer a certain ability to convene as well as legitimacy within the pig research and animal health sector worldwide. This facilitates participation by, and contributions from, scientists from all over the world in its biannual forums, which is not necessarily the case for other conferences, which tend to be more regional. In this sense, IPVS has produced spaces for joining together communities made up of pig producers, veterinarians, diagnosticians and scientists in order to build a platforms for establishing ties and communication between these various sectors, which means "meeting" to debate, exchange ideas, discuss and analyze common concerns, methods, progress and results of research or new scientific horizons, so it refers to institutionally consolidated communities, and theses social platforms serve as spaces to reaffirm scientific-communal attributes, they provide mechanisms for reaffirming and exploring new communities and collective identities [61] and they generate a sense of belonging to this or that scientific group. According to Blanco [62], the congregate communities in congress constitute a central part of the social and institutional structure of the academies, since they organize circuits of intellectual debate, promote the careers of their members and organizers and generate structures for

the exchange of knowledge for future research areas, which is why they also constitute observatories for detecting prominent patterns in the generation of knowledge. In this context, considering Granovetter [53], these kinds of social spaces can be defined as communities with weak social ties; therefore, it has a significant potential for social innovation and innovating the knowledge gathered. The community, under these conditions, could be said to be temporary and invented [63], given the need to generate such a space; but when it is generated as a relational dimension, with interactions and a variety of actors, it becomes a concrete fact in terms of social confluences, objectives and results obtained [64].

In the context of this analysis, these kinds of temporary communities drive a specific area of knowledge that is being innovative in different ways: pig health and production. Based on this, it was decided to track a social particle which is contained in each of the swine scientific research documents that were published in the selected IPVS proceedings [54–58], because these documents present two basic characteristics that are important for this study: (a) They constitute a parameter for the local–regional–institutional scientific concerns related to questions on swine production and health, which allowed for the gathering of a predetermined, not probabilistic, population, and convening a set of 5003 authors, who generated a volume of 4868 research papers on a range of swine-related topics, which made it possible to group them by region and nationality: Central Europe (Austria, Czech Republic, Poland, Slovenia, Croatia, Slovak Republic, Switzerland, Romania, Lithuania); Northern Europe (Denmark, UK, Sweden, Norway, Finland, Ireland); Southern Europe (Italy, Spain, Greece, Portugal, Serbia); Western Europe (Belgium, France, Netherlands, Germany); South America (Brazil, Argentina, Colombia, Venezuela, Chile, Peru); North America (Canada, USA, Mexico); East Asia (China, Japan, South Korea, Taiwan, Mongolia); Southeast Asia (Thailand, Malaysia, Philippines, Vietnam); and Africa (Kenya, South Africa, Nigeria); Australasia (Australia, New Zealand). (b) Each research work contains a basic social formula for association and participation (co-authorship) predetermined by the authors from their academies, institutions, resources, needs and professional training: the author–co-author dyad. We suggest that for these types of documents, trust is managed more strictly than for other types of collaborations (an academic course, a research project, etc.) where dealing with this social factor might be somewhat more relative. Collaboration in writing and presenting the results of scientific research in a specialized international forum could involve economic resources and ethical issues, and therefore have administrative and legal implications. Based on this, we assume that scientific collaboration is also a way of conceptualizing the manner in which community synergy has been transformed into scientific conglomerates, revealing strengths and weaknesses generated by the dynamics of association. It is clear that the complexity of scientific collaboration cannot be expressed solely in terms of co-participation and/or co-authorship in research work, except to a partial extent [65,66]. However, the co-authorship structure provides access to internal social qualities and attributes related to the ability to manage cohesion and the social weight of each participant [67,68], which can be associated with mechanisms of communicational openness or closedness.

In this direction, the existence and multiplicity of dyads implies patterns of association and filial complicity, wherein diverse social formulas (committees, teams, crews, etc.) are constantly interacting with one another: collaborating, exchanging and sharing a variety of resources (economic, human, intellectual, etc.). This confirms the community roots of the knowledge generated in these dyads and intersections, as well as association models determined by the particular social experiences of each researcher (depending on their resources, needs and objectives) [69]. By extrapolating the amount and diversity of the social ties, we can synthesize structural elements such as diversity, synergy, reciprocity, trust, affiliation, empathy, etc.

This relationship made it possible to define the unit of analysis to access the structural feedback loops of collaboration underlying the presentation of pig research work. Upon establishing this condition, all works submitted by a single author were excluded. Our

interest in the underlying social models present in the papers presented in the selected IPVS congresses led us to the proposal for using the social network approach.

2. Social network analysis. The social agglomeration that forms around specialized scientific knowledge involves individuals, links or ties, information flows and feedback. This, in turn, makes it possible to consider two central elements: complexity and topological dimensionality. In other words, a social event that can be singled out at a certain moment of its reproduction, which makes it possible to establish a variety of measurements and delimitations that will allow us, in specific circumstances, to conceptualize and measure the diameter of a social network structure. In this sense, the isomorphic behavior derived from a certain level of social wealth and complexity, consisting, in turn, of the reticular exponential behavior of multiple egocentricities joined together, reveals social behaviors of affiliation, preferences, empathy, complicity, etc. That is, community behaviors that suggest a specific socio-centric value, and therefore, the opportunity for scientific communities to have a measurable record and balance sheet of qualities such as efficiency, communications innovations, management of social input, diversification of communication channels and exchange of knowledge, among others.

For the above, mathematical equalities of centrality and grouping were used, namely, degree centrality, betweenness centrality, social density and cliques.

Degree centrality. According to Freeman's [70] conception, this notion of centrality refers to the sum of direct adjacencies that a specific node has in its social environment, obtaining the possibility of strategically accessing the flow of information and knowledge that runs through its social network, and also increasing the degree of susceptibility to it. In regard to the proposed theme, the number of co-authorships that each author has, would express his/her degree centrality. The mathematical notation that allows us to calculate it is:

$$d_i = \sum_{j \in \vee} Aij, \; \forall i \; \in \; \vee \tag{1}$$

where $d_i$ = degree centrality of the actor in question and *Aij* = the sum of the matrix that joins nodes "*i*" and "*j*".

Betweenness centrality. According to the structuralist conception of society, all of its members construct it (subjectively and objectively) through the multiple daily events that they generate, which implies constant and multiple linking. Therefore, each pair of actors will have the possibility of linking by way of more than one path, although one of these will be the shortest, most functional and most economical, and will go through a specific mediator. The frequency with which this node allows for the connection between these pairs is called the betweenness centrality and it expresses the ability to define the ties between actors and even determine a community's social cohesion [71,72]. Freeman [70] and Brandes [73] propose the following mathematical equation:

$$g_k = \sum_{i>k>j} \frac{g_{ikj}}{g_{ij}} \forall k \; \in \; \vee \tag{2}$$

where $g_k$ = degree of intermediation (betweenness); $g_{ij}$ = N° of geodetic distances between nodes "*i*" and "*j*"; and $g_{ikj}$ = N° of geodetic distances that exist between "*i*" and "*j*" and that pass through "k".

A network structure's transitive quality expresses its effectiveness in managing the flow (direction) and speed of the transfer of information between nodes. This quality is known as social density and is directly proportional to the multiplicity of the possible links among a certain group (real connections). Therefore, high social density values indicate greater structural efficiency in managing the social input. According to Wasserman and Faust [71], the values range from 0 (there are no ties) to 100% (all nodes are fully linked) and it is possible to calculate this with:

$$\Delta = \frac{L}{g \, (g-1)} \tag{3}$$

where $\Delta$ is the density; $L$ is the number of real relations; and $g\,(g-1)$ is the number of possible relationships.

The shared interests and values of empathy and social cohesion, among social actors, reveals a network's structural transitive capacity and also the dynamics of social overlap within it [74]. This allows one to identify dense, compact and connected groups called cliques. Significant overlapping values indicate greater exchanges of information between cliques, therefore, possible values of innovation or social recovery of the structure in question. Brandes and Erlebach [75] point out that Turán [76] defined the calculation to determine the presence of cliques of certain proportions, depending on the size of a network, as follows:

$$G = V,\ E\ if\ m > n^2/2 \cdot (k-2)/(k-1) \tag{4}$$

In this context, $G = (V, E)$ is an undirected graph, so there is a clique the size of k divided by $G$.

Using these categories, the structural centrality of the actors was accessed, based on the set of co-authors registered in the scientific works analyzed. Analysis with the notion of graphs allowed us to express the connections among researchers by congress, by region and by country. These graphs were made with the UCINET6 for Windows program, 6.587 [77].

In this regard, in gathering and compiling the information for this research, and extracting the agglomeration patterns that were of interest for us, one might suggest a certain approximation to the area of artificial intelligence (Machine learning) with notions such as "big data" and "data mining". However, in contrast to these tools, wherein the information is managed by automated systems on digital platforms that collect social information, the information analyzed here was extracted and captured directly from the IPVS forum documents, in order to respond to some basic social science questions: with whom, how, on what topic and where did the researcher carry out his/her work? This, in turn, allowed us to respond to the general inquiries necessary for our work. Furthermore, we believe that epistemological approaches combining tools such as machine learning and social sciences could mean a new multidisciplinary field, providing opportunities to designing new theoretical constructs and find new interpretations for social phenomena [78,79]. Additionally, we need to clarify that the study period (2010–2018) was selected due to the systemic social delimitations caused by the H1N1 influenza (Mexico, 2009) and COVID-19 (2020) pandemics, generating similar processes of social organization (decreased social mobility, health alerts for travelers, instructions about how to sneeze, use of antibacterial gel and face masks, as well as the first forms of social distancing), which, in the period analyzed, might have led to particular behaviors in the research structures related to the animal health and production sectors.

## 3. Results and Discussion

The community for research on pig production and veterinary health, by way of the various forums for discussion and analysis in which it has participated (IPVS 2010–2018) [54–58], given the recurrences and reciprocities expressed through the basic unit of analysis of this research (co-authorship), contained in the research works themselves, suggests a particular form of a communication ecosystem that is both broad, with a variety of behaviors, and complex based on the social hetero-reactivity that constitutes it (Figure 2).

This system, when dissected by each of the agglomerations analyzed, indicates different structural behaviors in terms of social bonding and management of the prominence of the actors, which is observed in the "local" behavior of each agglomeration considered. Although the influx of actors determines said behaviors, the structural configuration is maintained for each event, thus suggesting heterogeneous topological values. In this sense, we can observe a variety of social densities, different groupings, areas with greater social concentrations and areas with only limited ties to the universe analyzed.

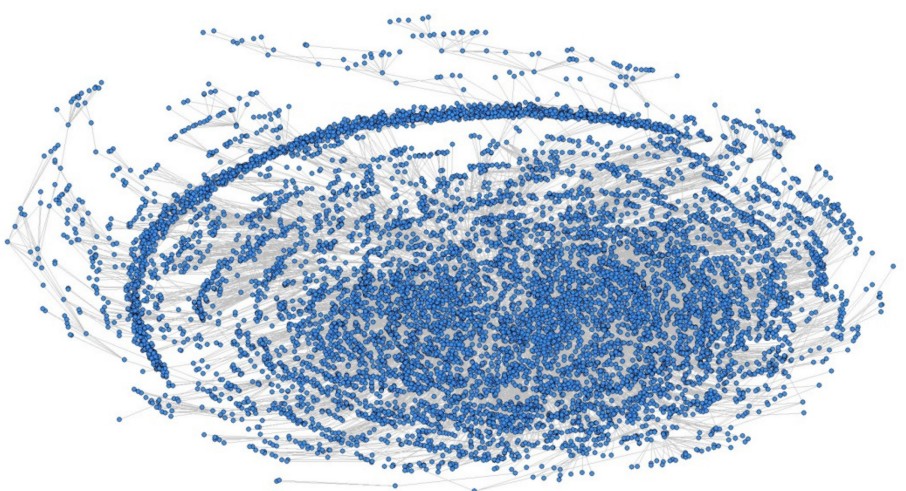

**Figure 2.** Structural behavior in a global scientific community of swine research. Source: [54–58].

*3.1. Research Structure-2010*

For this specific time and social space, 2010, the research structure for swine issues (production and veterinary health) exhibited social values as the next: the average for the degree centrality was 3.066 links, with a maximum of 53 social arcs, a variance of 8.68, and, nonetheless, a standard deviation of 4.12. This suggests a certain variation in the number of direct ties, and a more or less uniform distribution of social prominence based on the capacities to access the information circulating in the structure. When segmenting the degree centrality by strata, it was observed that 68.42% of the population analyzed established between 1 and 2 direct links and only 2.72% had direct links ≥15. This tendency is exacerbated when the betweenness centrality was analyzed.

In this regard, 97.39% of the intermediations were present between 1 to 50 links between actors, 1.83% of intermediations were present in from 51 to 300 links and 0.78% had values above 300 links. The presence of conventional cliques (made up of three actors) was 493; as the social composition of this type of group increased, only 93 were found. This evidenced a greater control of information flows and actors that centralized more information, which suggests a low social density and limited social overlap for this structure (Figure 3).

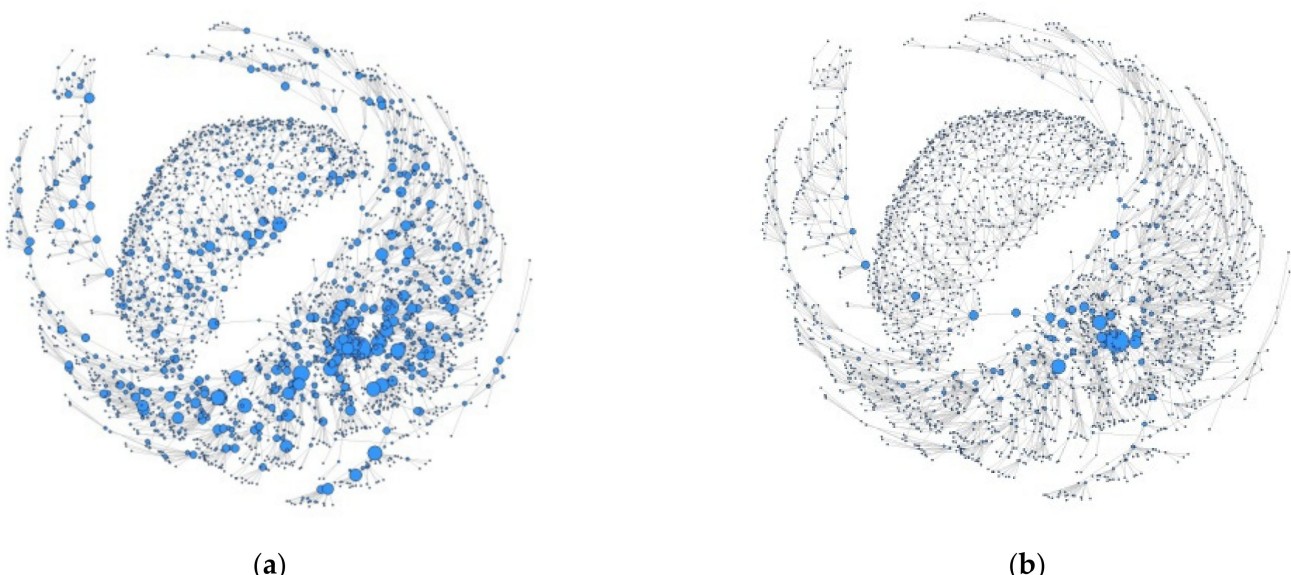

(**a**)          (**b**)

**Figure 3.** Measures of centrality (**a**) degree and (**b**) betweenness in pig research worldwide. Source: [54].

### 3.2. Research Structure-2012

In the social and spatial agglomeration for 2012, the research social structure presented lower values than for the preceding year (it is likely that in 2011 the structures for research on pig production and veterinary health were still feeling the impacts of the A H1N1 influenza epidemic in Mexico in early 2009, although there is no evidence to corroborate this). The average of degree centrality for this structure was 2.84 links, with a maximum of 61 social arcs, a variance of 12.99 and a standard deviation of 3.60. This points to greater variations in local values with respect to the average, but with a lower standard deviation than the preceding year. When analyzing the degree centrality by strata, 68.37% had 1 to 2 links, 30.24% had 3 to 14 direct links and 1.39% had ≥15 links pointing to a pyramidal structure of information management. These data correspond to ranges for betweenness centrality in which 78% of the social bridges allowed for from 1 to 50 direct linkage formulas, 15.46% facilitated between 51 to 300 links and 6.48% did so for >300 links. When calculating the number of cliques, there were 442 and by increasing the composition to 4 members, the number was reduced to 69 cliques (Figure 4).

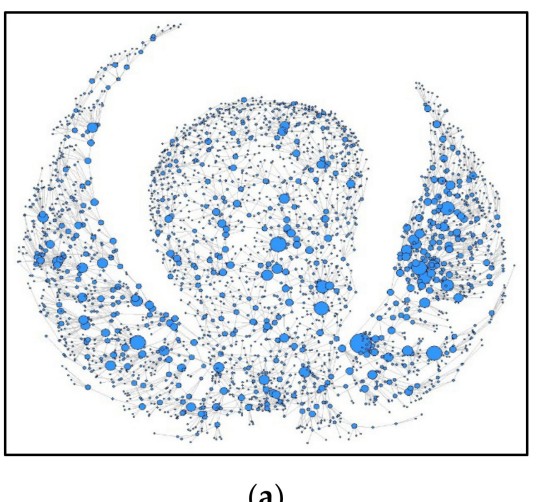

(**a**)

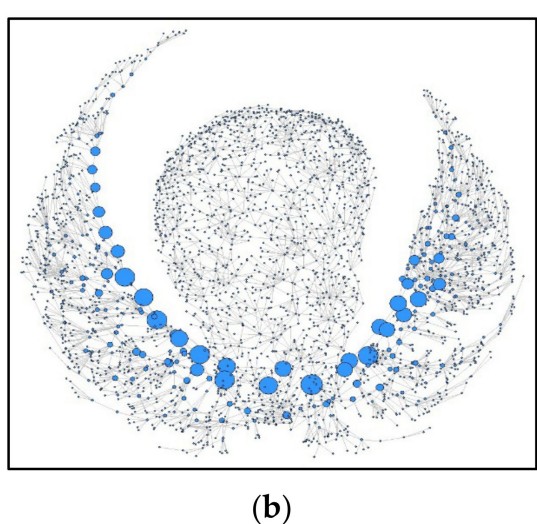

(**b**)

**Figure 4.** Measures of centrality (**a**) degree and (**b**) betweenness in pig research worldwide. Source: [55].

### 3.3. Research Structure-2014

The social structure corresponding to the year 2014 shows a rather close-knit set of values, moderately lower than those of the previous structure, with an average degree centrality of 2.83, with a maximum of 44 social arcs, a variance of 12.44 and a standard deviation of 3.52. This indicates a mean variability in the local values, with respect to the average, and a standard deviation which is lower than that of 2012. Furthermore, the stratified degree centrality indicated, as in the previous cases, a hierarchical social structure. In this sense, 69.09% of the scientific community analyzed established 1 to 2 links, 20.65% had 3 to 6 links, 8.72% had 7 to 14 links and only 1.5% of this population achieved ≥15 links. This polarization is confirmed when analyzing the value for betweenness centrality, which reported that only 3.54% of the population managed to act as a bridge node in values above 150; 9.57% of the individuals managed to link to subjects from their own network between 51 and 150 times; and 86.88% served as a bridge between 1 and 50 times. The number of cliques under the two previous conditions was 395 and 94 (Figure 5).

### 3.4. Research Structure-2016

In 2016, the scientific structure, specialized in swine issues (health and animal production), yielded an average of 2.79 links per actor, with a maximum value of 36 direct ties, a variance of 10.87 and a standard deviation of 3.29. This indicates a structure with little social variability. When addressing the degree centrality, in a stratified way, it was

observed that 68.8% of the actors had 1 to 2 direct links, 29.88% had between 3 and 14 links and 1.31% $\geq$15 direct ties. This hierarchy in communication is partially confirmed by the betweenness centrality, but atypically so, since although a decrease in this type of social prominence is observed in the values, for 1 to 50 times as a bridge node (76.11% of the population), from 51 to 101 (11.95%) and from 102 to 150 (2.39%), in the stratum of 151 to 200 participations as a bridge node, the value increases to 4.10%, although it decreases to 1.37% in values from 201 to 300 times and increases again to 4.10% in values greater than 300. The total number of working groups was 421 cliques (of 3 members) and 64 (composed of 4) (Figure 6).

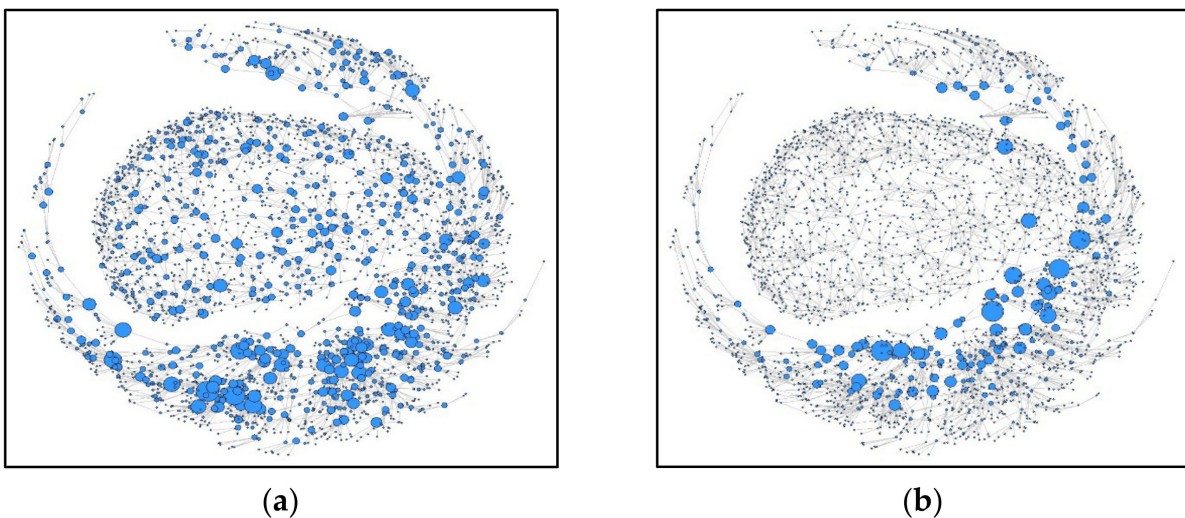

(**a**)  (**b**)

**Figure 5.** Measures of centrality (**a**) degree and (**b**) betweenness in pig research worldwide. Source: [56].

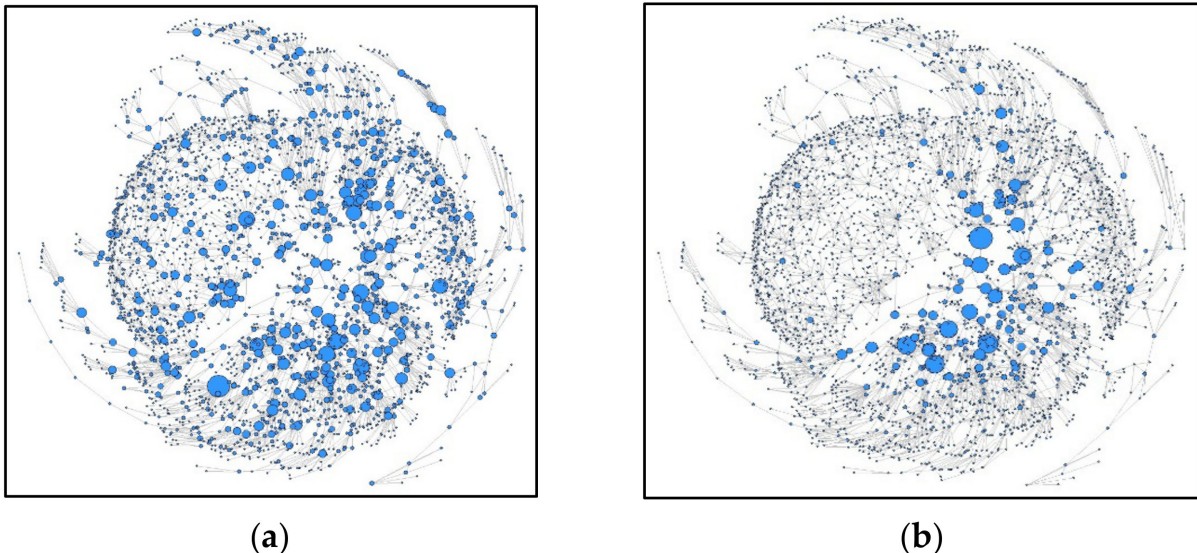

(**a**)  (**b**)

**Figure 6.** Measures of centrality (**a**) degree and (**b**) betweenness in pig research worldwide. Source: [57].

*3.5. Research Structure-2018*

In the last year analyzed, the structure had a conventional behavior in accordance with the previous models: concentration and hierarchization of social prominence. The measure of degree centrality, by strata, was concentrated in 1.64% of the actors (with values $\geq$ 15 links), 27.35% of the population had 3 to 14 links and 71.01% of the actors had 1 to 2 links. This indicates a concentration of the capacities for access to information and, henceforth, of the strategic position to modify information flows. This social prominence is

confirmed by the notion of power contained in the capacity for betweenness: 4.65% of the researchers stood out with values of >150 times as a bridge node, 11.16% of the population functioned 51 to 150 times as a bridge node and 84.19% between 1 and 50 times as an intermediary in the relationship of each ordered pair. The count for cliques of 3 members was 340, by increasing the number to 4, it was reduced to 65. The foregoing is indicative of significantly centralized structures, where only some researchers are concentrating social weight and, therefore, generating directed and/or limited flows of information (Figure 7).

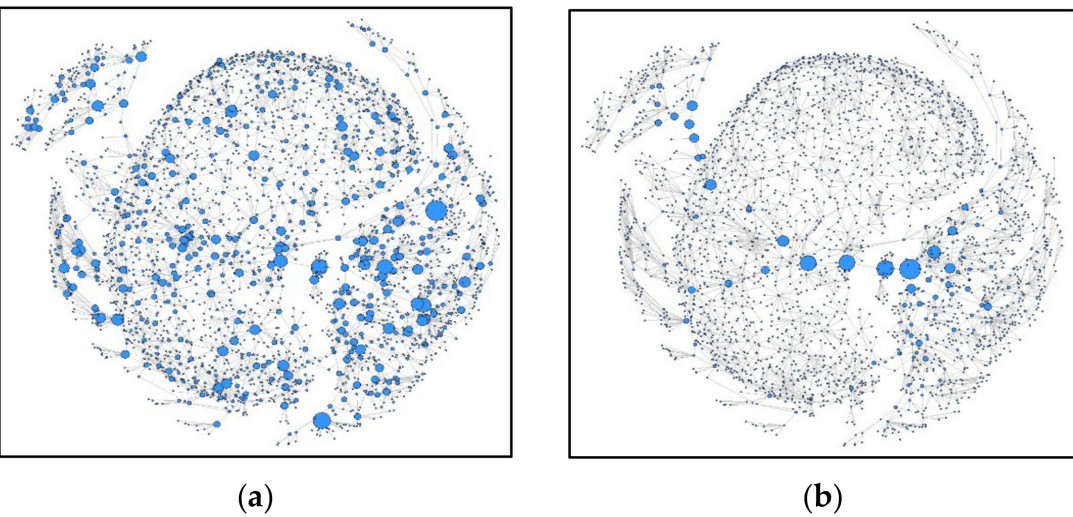

(**a**)                    (**b**)

**Figure 7.** Measures of centrality (**a**) degree and (**b**) betweenness in pig research worldwide. Source: [58].

All of the above is characteristic of a structure with a low average of degree centrality (2.85) and a betweenness centrality of 6.33, and, therefore, with limited possibilities for exhibiting horizontal behavior (considering the number of nodes involved: 5003), due to the concentration of resources and access to information. The average of betweenness centrality was just 0.02%. Such values indicate a centralized and hierarchical structure: little social variability in links and in the hub nodes for links (Figure 8). It is possible that the level of specialization of these groups is such that their dominance in the scientific-commercial market is forceful, but at the expense of nullifying their behavior as a social scientific system.

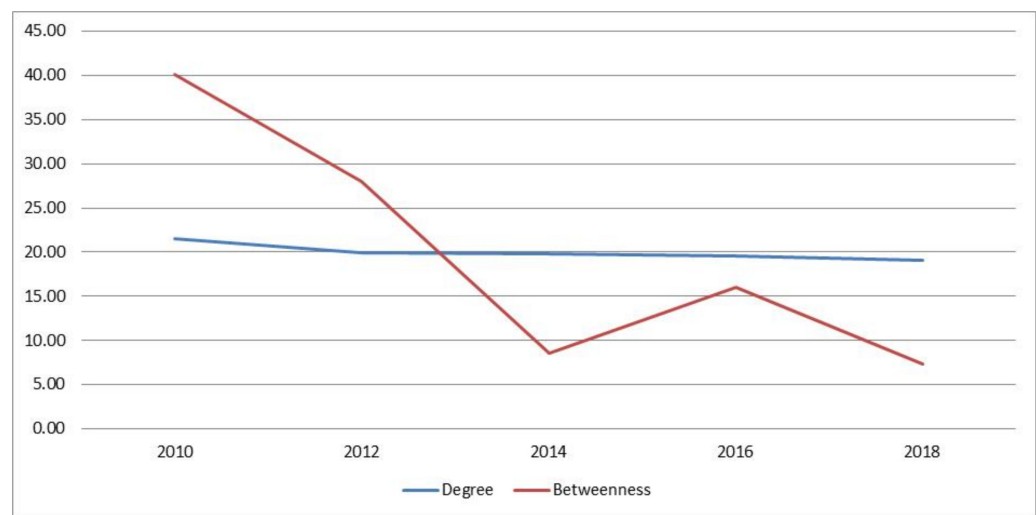

**Figure 8.** Average tendencies for degree and betweenness in a global scientific community of swine research. Source: [54–58].

### 3.6. Regional Structure of Pig Research 2010–2018

The formation and opening of economic blocs in the world, in the mid-1980s, further modified and dynamized local and regional livestock production systems that had already been growing as a result of secular changes (population increases, urbanization, economic growth and accelerated modification of diets, among others) in the final decades of the 20th century [10]. The constitution of economic blocs through free trade agreements was considered to be an appropriate mechanism for achieving regional integration and adjusting for certain economic and technological asymmetries, by implementing, for example, standardization of quality and production indices. However, this could not always be achieved due to the existing social and economic differences within and between regions [80–82]. Swine production, such as other livestock sectors, responded in a variety of differentiated ways to the increase and diversification of the demand for products with various technological innovations (in biology, genetics, chemistry, machinery, among others), intensification of production in regions throughout the world and integration of production chains in order to stimulate increased productivity [83] (Figure 9) and generate measures needed to adapt to various structural changes that have impacted the industry's development [84,85].

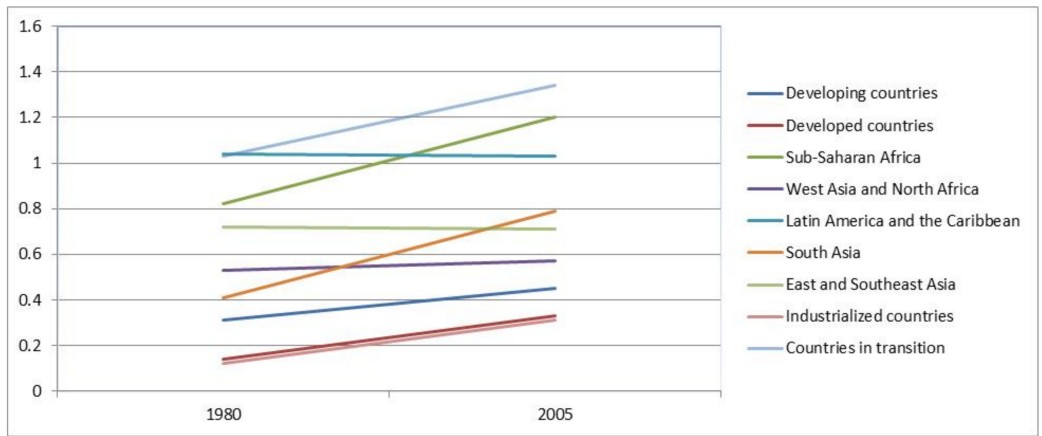

**Figure 9.** Pork productivity by world regions (kg of product/kg of biomass/year). Source: Elaborated with data from Steinfeld et al. [10].

This allows us to assume that the binominal pig research–production systems, which were structured regionally and even more specialized, modified their operating structures (mainly in veterinary and meat health), and acquired specific characteristics, but began to face unprecedented challenges due to the growth and uniformity of the pork sector's productivity worldwide [86–88]. Based on this, the social agglomeration of the research structures allows for their regional grouping in topological terms.

In this direction, it was observed that at the regional level a trend prevailed with low communication values (3.3), although with significant intermediation values (134.2). This suggests the presence of emerging social subsystems with autonomous capacities, although differentiated, in their collaborative processes, which brings to mind negotiated and moderately high patterns of network centralization.

When comparing the structural behavior of the regional systems (Table 1), we observed that those that presented the highest degree centrality values, Southern Europe and North America, showed a relative structural equivalence, due to their ability to communicate (degree centrality), although, according to Bonacich [89], this does not imply that they have the same social weight in terms of linking power (betweenness centrality). In this sense, while Southern Europe has a betweenness centrality of 328.731, North America has a betweenness centrality value of 531.72, which implies that information is much more diversified in this region. In turn, South America and Western Europe have similar nodal behaviors, although with quite a significant difference in the weight and particular quality of intermediation. For its part, Northern Europe has a degree centrality similar to East

Asia and Southeast Asia but somewhat higher than that of the latter region. In the case of Western, Southern and Northern Europe, these significant assessments could be explained by the European directive 1027 and the mandate issued by the European Commission in February 1994 (European System of Evaluation of Veterinary Training) which associates, on a voluntary basis, all European veterinary faculties that so wish, thereby promoting, developing and harmonizing veterinary education, and above all enhancing cooperation among faculties, mainly European ones, and also with other relevant organizations [90,91].

**Table 1.** Degree centrality and betweenness centrality (Averages) by world regions.

| Options | Degree Centrality | Betweenness Centrality |
|---|---|---|
| Southern Europe | 4.01 | 328.73 |
| North America | 3.97 | 531.72 |
| South America | 3.58 | 17.65 |
| Western Europe | 3.52 | 282.08 |
| Central Europe | 3.32 | 27.88 |
| Eastern Asia | 3.22 | 91.36 |
| Southeast Asia | 3.19 | 22.59 |
| Northern Europe | 3.13 | 39.20 |
| Australasia | 2.63 | 0.39 |
| Africa | 2.16 | 0.82 |

Source: [54–58].

With respect to other regions, similar behaviors are observed in the degree centrality values, but with extremely low betweenness centrality values, with the exception of East Asia, which could be associated with much more vertical systems in the distribution of social prominence. In this respect, if hog production systems are understood in regional terms, this indicates that there could be differences (among regions) in the capacity to manage and distribute information worldwide and, therefore, differences in the capacity to face systemic dilemmas associated with massive hog production.

*3.7. Pig Research Structure from 2010 to 2018 in Six Producer Countries*

Different outstanding features and structural behaviors can be observed in each regional research structure. Some commercial and technological factors that intervene here are the centralization of the production system, competition between regional systems and the geographic dispersion of the systems, as well as the presence of incipient systems. This, at the country level, has nodal repercussions since it is related to the commercial and technological competition that exists in the pork sector and is probably also affected by the trade–research binomial that drives these structures nationally. The great disparity in pig production in the world by country (Figure 10) suggests dynamic, hierarchical, commercially prominent, specialized, particularized, but also intricately diversified, research systems.

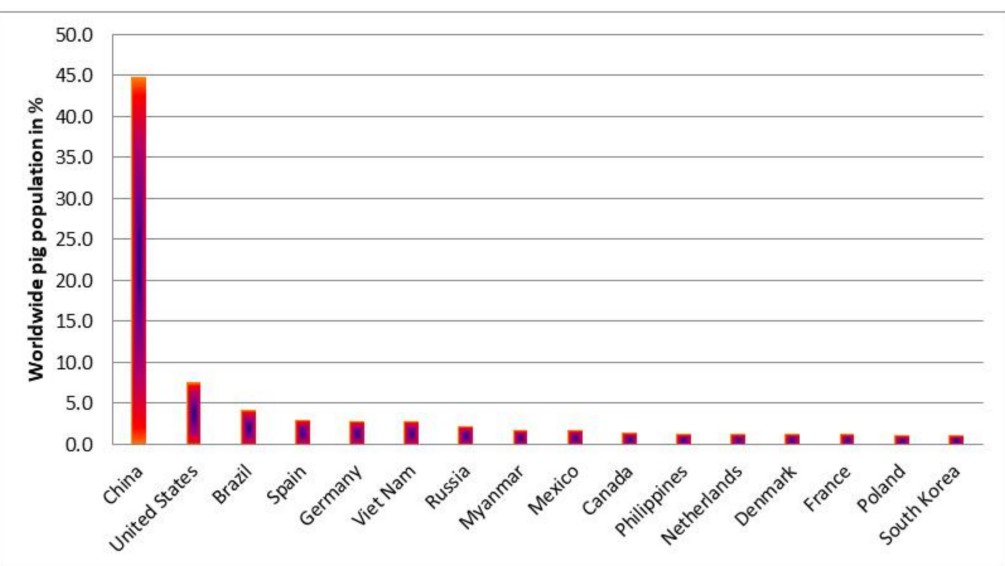

**Figure 10.** Distribution of the pig population in the world (main producers) (%). Source: Elaborated by the author with data from FAOSTAT [92].

An example of the above can be observed in the structural and productivity changes in pig systems during the 2000s as a result of the expansion of internal and international markets. In countries such as China, which represents around 45% of the world's pig population, the development of the global pork market led to a change from small-scale production to medium and large technologized farms. However, in the Chinese case, it is important to point out that their meat inventories show some inconsistencies: the meat supply reported is much larger than the demand; there is a system of socioeconomic incentives, within the government, to inflate production data; and there are contradictions in the calculations of losses, in general, and losses in the pig production chain, as well as in the data on pork meat consumption away from home. In addition to this, the consumption of pork meat in China's rural areas has been underestimated by about 30% [93,94].

The Republic of Korea adopted a system of technological innovation with the help of the state. In Latin America, domestic demand expanded and forced countries to make significant investments in this sector. In the United States and Canada, the pig sector was extensively restructured. These dynamics of restructuring, investment and increased productivity have taken place in various regions of the world [83], to the point that it is possible to characterize them based on the behavior of the volume of production, as well as its value (Figure 11).

These six countries were selected for our analysis because, in addition to being prominent actors in world swine production, they are considered as research hubs in clinical swine science and production, as well as being significant contributors to the number of research papers published in the IPVS proceedings (2010–2018). The USA and China stand out with 14.70% and 9.25%, respectively, of the total number of papers presented, as well as consolidated economies, such as Spain (7.94%), South Korea (7.03%), Germany (6.39%) and Mexico (5.82%). The latter as a Latin American economy in the process of economic consolidation. These six countries account for 51% of the total research carried out on pig health and production in the period of interest for our research (Figure 12).

In this context, the productive capacity of each country has been modified as a result of its participation in the competition for pork markets in recent decades. We can assume, therefore, that the market evolution of the pig sector is directly related to the specialization and expansion of research structures in this sector. The market has been the main driving force for this. Consequently, it seems that each social research structure in the pig sector, in the areas of health and production, adopted certain specific patterns of social management.

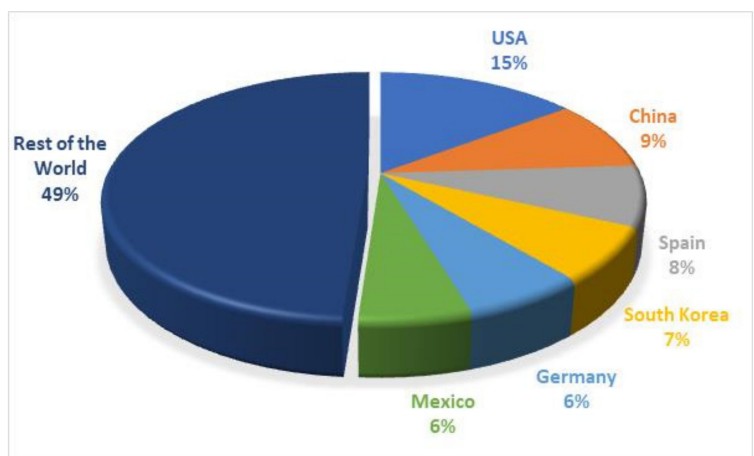

**Figure 11.** Distribution of the value and volume of pig production in 6 countries 2000–2019: (**a**) USA, (**b**) China, (**c**) Spain, (**d**) South Korea, (**e**) Germany and (**f**) Mexico. Source: FOASTAT [1].

**Figure 12.** Top six research countries. Source: [54–58].

As a result, we can see that the main producers in the world, the USA and China, developed diversified research networks, based on being able to manage their economic prominence, and also given the dimensions of their pig production systems. This also holds for other research systems. For example, in terms of communication capacity and

access to information (degree centrality), in descending order, Spain and the USA exhibited the best communication values and they maintained a certain social prominence in these two aspects; a second group is composed of South Korea and Mexico and a third group with Germany and China. Regarding the quality of betweenness centrality, the USA has a great capacity for linking and exchanging information among cliques of researchers, and therefore, greater social overlap within its research networks, followed by South Korea, Spain (with a more vertical social imbrication), Germany (the strengths of economically consolidated countries such as the USA and Germany stand out, although the latter presented lower values in terms of its research networks), Mexico and China in last place. In terms of the reciprocal exchange of information, we observed that South Korea has a highly inter-communicated network, followed, in descending order, by the USA, Spain, Mexico, China and Germany. The latter is noteworthy, since in terms of production volume, it occupies third place worldwide. On the other hand, with regard to the effectiveness of information management (social density), the Mexican research network, characterized by a high level of social overlap, showed the best performance, followed by South Korea, Spain, Germany, the USA and China.

The USA had the most significant values, which indicates a more interconnected and diversified network in terms of bridge actors, but it also means that some information may be tainted by inputs from the bridge actors. South Korea and Spain would represent, under this assumption, a medium model of betweenness. In the case of Germany, its betweenness centrality would be moderate, despite being a prominent player in world pig production. For its part, Mexico has a low value in this rating, which implies (considering its participation in the pig health and production markets) a limited body of research, although with clearer structuring qualities. It is noteworthy that there are some notable aspects for countries such as Spain and Mexico, which have economies of less regional weight and are in the process of consolidation. Spain maintained significant structural values and Mexico stood out for the effectiveness of its network structure for managing information (which may be due to the fact that it is a much smaller and thus more manageable network of researchers). It should be clarified that in the case of the low values that China exhibits (in spite of occupying second place in terms of social overlap), we suggest that, in addition to having a highly centralized bureaucracy, the observed "lack of sociability" may be a result of the fact that the pig farming system that it must manage is gigantic, which leaves only a very narrow margin for social innovation (it owns 45% of the world's herds) [84], so China's structural contribution was the most unusual and limited. In general, the information exchange processes were markedly vertical; by varying the composition of the work groups from three components to four, the result was an average decline in social overlap to just 27.34% of the preceding figure (Table 2).

**Table 2.** Descriptive statistics for six countries participating in IPVS 2010–2018.

| Options | Degree | Betweenness | Social Density (%) | Reciprocity (%) | Cliques (3) | Cliques (4) |
|---|---|---|---|---|---|---|
| USA | 3.99 | 444.22 | 0.15 | 6.16 | 374 | 103 |
| South Korea | 3.90 | 280.51 | 0.28 | 9.24 | 214 | 54 |
| Spain | 4.14 | 222.33 | 0.26 | 4.57 | 243 | 77 |
| Germany | 3.12 | 122.79 | 0.21 | 2.65 | 132 | 21 |
| México | 3.85 | 96.75 | 0.31 | 4.54 | 160 | 42 |
| China | 2.85 | 3.52 | 0.11 | 3.72 | 256 | 66 |

Source: [54–58].

## 4. Conclusions

A social network approach made possible the access to a partial comprehension of scientific conglomerates which are specialized in health and swine production. The analysis focused on the co-participant scientific production which unveiled several social structures involved in the communication and exchange of some information inputs. In consequence,

different structural qualities from worldwide swine research were confirmed. In this direction, the adjacency dynamics characterization has permitted the determination of the underlying pattern of the growing centralization of social prominence at the swine research systems, so technological and scientific areas are remarkably hierarchized but isolated and thus prone to poor communication and poor information sharing. Therefore, there are areas with homogeneous and wider technological habitats, but epistemologically disconnected in a global agri-food sector which is increasingly analogous and massive. Consequently, there are some different strengths and weaknesses in each regional and national swine research subsystem: (a) groups with social prominence in the access of flows of information, skills connection within scientific communities and the quality of the exchange of information; (b) the strengths of economically consolidated regions although with lower values in terms of its research networks; (c) regions which have economies of less regional weight but with significant values in the effectiveness of its structure in network for managing information; and (d) the structural challenge of reorienting the production system from local farms to mass production.

The control of the agri-food narratives goes through a different structural sieve: there are scientific communities and groups that exhibit the most solid scientific, social and commercial structures. However, in the analysis carried out, there were different qualities observed in the efficacies of these systems for constructing and exchanging scientific and technological knowledge. In a similar fashion, the social values obtained suggest structures characterized by limited and partial information flows, with low densities and with few possibilities for interconnection and collaboration between researchers. By having so few actors with high centrality values, these structures have greater control (and restricted social innovation) over the flow of information and communication within the social structure. This social marked hierarchy means that there is a high probability of disabled and inadequately re-channeled information flows as well as the presence of isolated groups in the networks. This means there are social structures with a high probability of collapsing if the central actors are removed, therefore with little systemic capacity for recovering from de-structuring events (e.g., biosafety gaps). Our reasoning is that these types of social structures are involved in strategic sectors (agri-food) in which the omission of updating sensitive information can give rise to socially disastrous scenarios (e.g., COVID-19 pandemic). Derived from these assumptions, we propose that supranational organizations, which are considered as global and international agri-food regulators such as the FAO, PNUD and WHO, among others, should be monitoring said behaviors in order to be able to assist states and corporations in the generation of standards for the interaction of research structures in this kind of sector, in order to maintain the necessary social quality of information and knowledge. The emergence of pandemics seems to indicate a plethora, throughout the world, of closed systems, that are poorly communicated and with little willingness to share information. These lead us to wonder if science has backed down from its own social responsibility by subjecting itself to criteria of political and commercial convenience (the regional hoarding of COVID-19 vaccines indicated evidence in this regard). Finally, it is necessary to point out that this analysis is presented as a general perspective for analyzing these types of issues based on the approach proposed. It does not address all of the social complexities of collaboration structures among researchers, which means that it is also necessary to explore these phenomena in terms of other types of solidarities and from different epistemological angles.

**Author Contributions:** Conceptualization, J.F.N.-E.; methodology, J.F.N.-E., F.E.M.-C. and F.Á.-P.; formal analysis, J.F.N.-E., F.E.M.-C. and M.C.R.-R.; investigation, J.F.N.-E.; data curation, F.E.M.-C. and F.Á.-P.; writing—original draft preparation, J.F.N.-E.; writing—review and editing, F.E.M.-C.; visualization, J.F.N.-E. and M.C.R.-R. All authors have read and agreed to the published version of the manuscript.

**Funding:** This research received no external funding.

**Institutional Review Board Statement:** Not applicable.

**Informed Consent Statement:** Not applicable.

**Data Availability Statement:** All data used in this manuscript were obtained from the proceedings of five world congresses (2010–2018) held by the International Pig Veterinary Society (IPVS), which are public access documents too: http://www.theipvs.com/links/ (accessed on 5 May 2020).

**Conflicts of Interest:** The authors declare no conflict of interest.

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
