# Peer review of "A Structural Approach to Some Contradictions in Worldwide Swine Production and Health Research"

_sustainability, doi:10.3390/su14084748_

Round 1

Reviewer 1 Report

This paper uses a network analysis to study the structure of IPVS. General methods are ok; the topic is interesting, but the writing could be improved.

1, The title of paper talks about paradox. However, I do not find any paradox. The title should be changed.

2, The section "Materials and Methods" could be separated.

3, The writing could be intensively improved. There are so many spelling and grammatic errors. 

Here are a few.  I can not list all.

e.g. 

line 25: . It was possible (to) analyze swine research dynamics, such as convergence and influence,

line 328:

Where D(d)i = Degree centrality of the actor in question, and Aij = the sum of the matrix that joins nodes “i” and “J” (j).

4, Fig. 10 talks about the pig statistics. It is well known there are huge statistical errors at least found in China (e.g. Yu and Abler 2014 & 2016). Some discussions are necessary.

1),  Yu X.  and D. Abler (2014). “Where Have All the Pigs Gone? Inconsistencies in Pork Statistics in China”. China Economic Review, Vol.30: 469-484.

2)  Yu X. and D. Abler (2016) “Matching Food with Mouths: A Statistical Explanation to the Abnormal Decline of Per Capita Food Consumption in Rural China”, Food Policy. Vol.63:36-43.

5, This papers talks about convergence and machine learning. In the literature, there are some literature related to club convergence , and clusters which is a typical non supervised machine learning. Some discussions and comparisons could be helpful to show the advantages of this study.

1) Zhang X. et al.2020Wheat yield convergence and its driving factors in countries along the Belt and Road. Ecosystem Health and Sustainability. Vol. 6.1819168.

2) Tian X. et al. (2019) Crop Yield Gap and Convergence in African Countries. Food Security. Vol.11:1305–1319.

3) Graskemper et al.  (2021). Farmer Typology and Implications for Policy Design – an Unsupervised Machine Learning Approach.   Land Use Policy.

4)Graskemper et al.  (2021) Values of farmers: Evidence from Germany. Journal of rural studies.

I believe this paper could contribute to the literature after taking into account these comments.

Author Response

This paper uses a network analysis to study the structure of IPVS. General methods are ok; the topic is interesting, but the writing could be improved.

Answer: The writing was verified

Comments and suggestions 1, The title of paper talks about paradox. However, I do not find any paradox. The title should be changed.

Answer: New title: “A Structural Approach to some Contradictions in Worldwide Swine Research

Comments and suggestions 2, The section "Materials and Methods" could be separated.

Answer: It was considered as a recommendation so it was not implemented because it would affect the narrative of the paper.

Comments and suggestions 3, The writing could be intensively improved. There are so many spelling and grammatic errors.

Here are a few.  I can not list all.

Answer: it was corrected

Comments and suggestions line 25: . It was possible (to) analyze swine research dynamics, such as convergence and influence,

line 328: Answer: it was corrected

Comments and suggestions  Where D(d)i = Degree centrality of the actor in question, and Aij = the sum of the matrix that joins nodes “i” and “J” (j).

Answer: it was corrected

Comments and suggestions 4, Fig. 10 talks about the pig statistics. It is well known there are huge statistical errors at least found in China (e.g. Yu and Abler 2014 & 2016). Some discussions are necessary.

1),  Yu X.  and D. Abler (2014). “Where Have All the Pigs Gone? Inconsistencies in Pork Statistics in China”. China Economic Review, Vol.30: 469-484.

2)  Yu X. and D. Abler (2016) “Matching Food with Mouths: A Statistical Explanation to the Abnormal Decline of Per Capita Food Consumption in Rural China”, Food Policy. Vol.63:36-43.

Answer: it was added the next text (page 16) and it´s bibliography:

“However, in the Chinese case it is important to point out that their meat inventories show some inconsistencies: the meat supply reported is much larger that the demand; there is a system of socioeconomic incentives, within the government, to inflate production data; there are contradictions in the calculations of losses, in general, and losses in the pig production chain, as well as in the data on pork meat consumption away from home. In addition to this the consumption of pork meat in China’s rural areas has been underestimated by about 30%.”

86           Yu X.  and D. Abler. Where Have All the Pigs Gone? Inconsistencies in Pork Statistics in China. China Economic Review, 2014, 30: 469-484, doi 10.1016/j.chieco.2014.03.004

87           Yu X. and D. Abler. Matching Food with Mouths: A Statistical Explanation to the Abnormal Decline of Per Capita Food Consumption in Rural China, Food Policy. 2016,63:36-43. doi: doi.org/10.1016/j.foodpol.2016.06.009

Comments and suggestions  5, This papers talks about convergence and machine learning. In the literature, there are some literature related to club convergence , and clusters which is a typical non supervised machine learning. Some discussions and comparisons could be helpful to show the advantages of this study.

Answer: It was added the next text (page 9) and it´s bibliography:

“In this regard, in gathering and compiling the information for this research, and ex-tracting the agglomeration patterns that were of interest for us, might suggest a certain approximation to the area of Artificial Intelligence (Machine learning) with notions like “big data” and “data mining”. However, in contrast to these tools, wherein the infor-mation is managed by automated systems on digital platforms that collect social infor-mation, the information analyzed here was extracted and captured directly from the IPVS forum documents, in order to respond to some basic social science questions: with whom, how, on what topic, and where did the researcher carry out his/her work? This, in turn, allowed us to respond to the general inquiries necessary for our work. Furthermore, we believe that epistemological approaches combining tools such as Machine learning and Social Sciences could mean a new multidisciplinary field, providing opportunities to de-signing new theoretical constructs and find new interpretations for social phenomena [71, 72].”

71           Meneses R. M. E. Grandes datos, grandes desafíos para las ciencias sociales. Revista Mexicana de Sociología, 2018, 80, 415-444, doi://dx.doi.org/10.22201/iis.01882503p.2018.2.57723

72           Ankeny, R., y Leonelli S. Repertoires: A post-Kuhnian perspective on scientific change and collaborative research. Studies in History and Philosophy of Science Part A . 2016, 60: 18-28. doi: j.shpsa.2016.08.003

Reviewer 2 Report

The paper is quite interesting referring to the paradox of swine research. It is well written however there are suggestions to the authors as follows:

1) General comment: Maybe the authors should also take into consideration regarding their results and conclusions that IPVS's may be influenced by the fact that during the last decade the yearly organized ESPHM's became very popular for researchers, at least from Europe (mainly for proximity and economical reasons) and thus more preferable by many of them comparing to IPVS (at least the years that was held in another continent (2016 the 2 conferences were co-organized, Dublin, Ireland). Meanwhile, similar important congresses are organized regularly at other continents eg APVS.

2) page 3, 1st paragraph: It would be better if they could provide more recent literature instead of no 18

3) line 235: Please correct the italics at "2) The.."

4) line 243: Please consider changing the word "scientists" with the word "researchers" since some of the before mentioned are also scientists

5) line 251 and elsewhere: Please change to "academia or academics"

6) line 630: Please change to "German"

7) Page 17, paragraph 2: I suggest considering that biosecurity gaps in pig production are also closely linked with political decisions and regional laws implementation eg differences between European Union laws and other countries

8) Page 18, paragraph 2-3: I suggest to the authors to be less strict to their conclusions and suggestions since they based their results on a limited number of congresses while at the same period (2010-2019) high valued similar events are organized regularly in different continents. Similarly, they didn't take into consideration the importance of Internet communication on researchers' communication. Thus, it is not easy to evaluate all the relative parameters snd consequently the relative conclusions should be more cautious. 

9) Page 19, paragraph 1: I would like to kindly ask the authors to more extensively explain why and based on which evidence they are referring to and even more important why they link examples of socially disastrous scenarios eg  SARS-Covid19 pandemics and Libanon with the degree of communication and exchange of information between research networks in the areas of pig health and production.

10) General: I would like to kindly ask the authors if they took into consideration the existence or not of collaborations between researchers from different countries in each work or did they categorize each paper based on the nationality of the main author.

11) Page 19, lines 782-788: The authors based on very risky assumptions they propose international organizations, such as FAO to be involved in monitoring -and not only- the interaction of research structures in pig health and production since they conclude that their majority are closed systems that poorly communicate and they have a little willingness to share information.

Surprisingly, since they did not take into consideration in their methodology other forms of communication between researchers in the prementioned scientific fields they should, at least, mention them in the discussion sector as gaps of the paper. On the contrary, they proceed on biased conclusions based on a limited source of information (4 IPVS) and one could support that some of these conclusions could be considered offensive without sufficient scientific proof.

Author Response

The paper is quite interesting referring to the paradox of swine research. It is well written however there are suggestions to the authors as follows:

Comments and suggestions  1) General comment: Maybe the authors should also take into consideration regarding their results and conclusions that IPVS's may be influenced by the fact that during the last decade the yearly organized ESPHM's became very popular for researchers, at least from Europe (mainly for proximity and economical reasons) and thus more preferable by many of them comparing to IPVS (at least the years that was held in another continent (2016 the 2 conferences were co-organized, Dublin, Ireland). Meanwhile, similar important congresses are organized regularly at other continents eg APVS.

Answer: it was added the next text (Page 6):

“It should be pointed out that while this is not the only forum that provides a space for gathering information, analysis and data on swine research, its history as an organization allows us to infer a certain ability to convene as well as legitimacy within the pig research and animal health sector worldwide. This facilitates participation by, and contributions from, scientists from all over the world in its biannual forums, which is not necessarily the case for other conferences, which tend to be more regional”

Comments and suggestions  2) page 3, 1st paragraph: It would be better if they could provide more recent literature instead of no 18

It was added the next (page 3):

“The total nationwide cost of productivity losses in breeding herds and growing pigs in the US, due to PRRS, has been estimated at $664 million dollars (US) per year, which is a sizeable increase from the $560 million (US dollars) estimated in 2005. The 2011 study differed most significantly from the 2005 study in terms of the distribution of the losses between breeding herds and growing pigs. Losses in breeding herds accounted for 12% of the total cost of PRRS in the 2005 study, compared to 45% in the current analysis”

It was removed  18.       Neumann, E.J.; Kliebenstein, J.B.; Johnson, C.D.; Mabry, J.W.; Bush, E.J.; Seitzinger, A.H.; Green, A.L.; Zimmerman, J.J. Assessment of the Economic Impact of Porcine Reproductive and Respiratory Syndrome on Swine Production in the United States. J. Am. Vet. Med. Assoc. 2005, 227, 385–392, doi:10.2460/javma.2005.227.385.

Instead it was included: Holtkamp D.J., Kliebenstein J.B., Neumann E.J., Zimmerm J.J., Rotto H.F., Yoder T.K., Wang Ch., Yeske P. E., Mowrer Ch. L., Haley Ch. A.  Assessment of the economic impact of porcine reproductive and respiratory syndrome virus on United States pork producers. J Swine Health Prod. 2013;21(2), 72-84. Available online: https://www.aasv.org/shap/issues/v21n2/v21n2p72.html

Comments and suggestions  3) line 235: Please correct the italics at "2) The.." it was corrected

Comments and suggestions  4) line 243: Please consider changing the word "scientists" with the word "researchers" since some of the before mentioned are also scientists     it was corrected

Comments and suggestions  5) line 251 and elsewhere: Please change to "academia or academies     it was corrected

Comments and suggestions  6) line 630: Please change to "German"    it was corrected

Comments and suggestions  7) Page 17, paragraph 2: I suggest considering that biosecurity gaps in pig production are also closely linked with political decisions and regional laws implementation eg differences between European Union laws and other countries

Answer: it was included (page 18):

“Although biosecurity, in fields such as those mentioned, is a technological issue with many gray areas –since constraints such as: banking, political decisions and the imple-mentation of regional laws, commercial and industrial secrets, copyright protection, etc”

Comments and suggestions  8) Page 18, paragraph 2-3: I suggest to the authors to be less strict to their conclusions and suggestions since they based their results on a limited number of congresses while at the same period (2010-2019) high valued similar events are organized regularly in different continents. Similarly, they didn't take into consideration the importance of Internet communication on researchers' communication. Thus, it is not easy to evaluate all the relative parameters and consequently the relative conclusions should be more cautious.  

Answer:

  1. We indicated at the text about the works and about the subject population: “The subject matter presented was analyzed from a mixed approach… [the works] constitute a parameter for the local-regional-institutional scientific concerns related to questions of swine production-health, which allowed for gathering a predetermined, not probabilistic, population…” this permitted us an access to a volume of 4,868 research papers written by 5,003 authors.
  2. We underlined the next too: “given the limited scope of this paper, we are not aiming to analyze the global research system. It is not possible for us to consider “all” of the already existing research nor that which is currently underway, except with a very delimited, synthesized and specified exercise. .. we intend to raise issues and questions that need to be explored as part of such a process. That is why we are proposing the methodological approach we present here”.
  3. It was included the next text (page 7) “ We suggest that for these types of documents trust is managed more strictly than for other types of collaborations (an academic course, a research project, etc.) where dealing with this social factor might be somewhat more relative. Collaboration in writing and presenting the results of scientific research in a specialized international forum could involve economic resources and ethical issues, and therefore have administrative and legal implications.”
  4. It was included the next text (page 9): “Also, we need to clarify that the study period (2010-2018) was selected due to the systemic social delimitations caused by the H1N1 influenza (Mexico, 2009) and COVID-19 (2020) pandemics, generating similar processes of social organization (decreased social mobility, health alerts for travelers, instructions about how to sneeze, use of antibacterial gel and face masks, as well as the first forms of social distancing), which, in the period analyzed, might have led to particular behaviors in the research structures related to the animal health and production sectors”

Comments and suggestions  9) Page 19, paragraph 1: I would like to kindly ask the authors to more extensively explain why and based on which evidence they are referring to and even more important why they link examples of socially disastrous scenarios eg  SARS-Covid19 pandemics and Libanon with the degree of communication and exchange of information between research networks in the areas of pig health and production.

Answer. The example of Libano was removed.

Answer: It was included the next text (page 20):

This means social structures with a high probability of collapsing if the central actors are removed, which is part of the paradox.

A possible explanation would the next: (page. 20):

“This suggests that biosafety gaps are also associated with social groups characterized by limited and partial information flows, with low densities and with few possibilities for interconnection and collaboration between researchers who do not know each other. By having so few actors with high betweennesss centrality values, these have greater control over the flow of information and communication within the social structure. This marked hierarchy of a few actors means there is a high probability of disabled and re-channeled information flows; therefore, collaboration between researchers is subject to purely commercial criteria. Hence the significant hierarchy observed in the centrality categories, since having so few bridge actors generates isolated and semi-isolated groups in the network. This means social structures with a high probability of collapsing if the central actors are removed…”

Comments and suggestions  10) General: I would like to kindly ask the authors if they took into consideration the existence or not of collaborations between researchers from different countries in each work or did they categorize each paper based on the nationality of the main author.

Answer: To this work this methodological angle it was not develop because the different origins and mobility (among nations and enterprises) that each researcher reported year by year. So the nationality reported to each work came from the first author.

Comments and suggestions  11) Page 19, lines 782-788: The authors based on very risky assumptions they propose international organizations, such as FAO to be involved in monitoring -and not only- the interaction of research structures in pig health and production since they conclude that their majority are closed systems that poorly communicate and they have a little willingness to share information.

It was added the next text (page 20):

“Our reasoning is that this type of social system is involved in strategic sectors (agri-food) in which the omission and/or lack of updating of sensitive information can, paradoxically, give rise to socially disastrous scenarios (such as the covid-19 pandemic itself).  Based on these assumptions, we propose that supranational organizations, which are considered as global and international agri-food regulators such as the FAO,  PNUD, and WHO, among others, should be monitoring said behaviors in order to be able to assist States and corporations in the regulation and/or generation of indexes and standards for the interaction of research structures in sensitive sectors”

Comments and suggestions. Surprisingly, since they did not take into consideration in their methodology other forms of communication between researchers in the prementioned scientific fields they should, at least, mention them in the discussion sector as gaps of the paper. On the contrary, they proceed on biased conclusions based on a limited source of information (4 IPVS) and one could support that some of these conclusions could be considered offensive without sufficient scientific proof.

Answer: it was added the next text (page 20):

“Finally, it is necessary to point out that this analysis is presented as a general per-spective for analyzing these types of issues based on the approach proposed. It does not address all of the social complexities of collaboration structures among researchers, which means that it is also necessary to explore these phenomena in terms of other types of solidarities and from different epistemological angles.”

Reviewer 3 Report

The research offers a very practical contribution and investigates the presence of possible Biosafety gaps with the aim of seeking solutions.
The paper is developed in depth and clearly representing different scenarios and structural profiles in the face of weaknesses.
An important and original element is the fact that the authors operate a parallelism with the pandemic Covid-19.

Author Response

There were not suggestions from the Reviewer 3

Round 2

Reviewer 2 Report

The authors proceeded to several corrections and clarifications in this revised version. Nevertheless, my last comment in the first review "Surprisingly, since they did not take into consideration in their methodology other forms of communication between researchers in the prementioned scientific fields they should, at least, mention them in the discussion sector as gaps of the paper. On the contrary, they proceed on biased conclusions based on a limited source of information (4 IPVS) and one could support that some of these conclusions could be considered offensive without sufficient scientific proof" cannot be covered by these changes.

Author Response

The following corrections were made

Reviewer:

English language and style are fine/minor spell check required

Author:

Spelling checked (North American style)

Reviewer:

Are the arguments and discussion of findings coherent, balanced and compelling?: Can be improved

"Surprisingly, since they did not take into consideration in their methodology other forms of communication between researchers in the prementioned scientific fields they should, at least, mention them in the discussion sector as gaps of the paper. On the contrary, they proceed on biased conclusions based on a limited source of information (4 IPVS) and one could support that some of these conclusions could be considered offensive without sufficient scientific proof"

Author:

The following arguments were added at the text:

Pgs. 4-5:

Furthermore, such underlying structure can be observed –and measured- from the dynamics of scientific agglomeration and collaboration which offers analytical elements to understand social behavior from both technicians and academic sector which are part of those overlapping structures that confluent in the strategy for improved and increased swine production.

Pgs. 5-6:

It is possible to visualize one of these structures through collaboration via scientific activities between specialized human resources (technicians, consultants, researchers, etc.); for instance, research projects, books, publications in peer-reviewed journals or in conference papers, among others.  Most of these products contain a specific particularity: they were written by more than just one author, so this partnership allows understanding part of the complexity of one of the most intelligible and well-known forms of collaboration structures because the co-authorship defines a coherent and reasonable social scientific formula: confidence is essential at the moment of writing together a scientific paper [45]. And the co-author networks have become much more complex and wider than before, so it is possible to suggest organizational patterns by scientific areas within such networks therefore it is legible to suggest communities and clustering dynamics of the knowledge by scientific field 46]. In this context, the majority of the network analysis about co-authorship has been related to the writing (and citation) of books and publications in peer-reviewed journals [47, 48, 49]; however, there is a scientific collaboration form that has not been commented enough or at least as much as the latest: the co-authorship in conference papers. The participation in any conference, international congress o local forums, etc., begins with some rational, strategic and specific ideas: the field of knowledge to participate, the kind of ideas and research results to share and in what group it is possible to participate [50] and all of this because each conference could provide some opportunities to start building a scientific network, develop the science communication skills (advocating for your research field) and getting benefits like in-crease the visibility of your research, project collaboration, possible access to research funds, professional transitions and learn valuable information from others researchers working with common research objectives, and the most important this would foster friendship and confidence with others similar researchers) [51, 52]. 

Besides, each congress or conference is not just an academic event but it is a social, economic and even a political meeting as well; it even could be seen as an interstice dialogue between communities to facilitate political distension among people and institutions. We suggest that there are much more possibilities to fertilize the knowledge because of the temporal quality of the communities at conferences especially because the weak ties system that prevalence in there [62]. In the conferences the scientific communities explore new research resources and social bonds and given the size of the academic agglomerations it is possible to identify social and massive movements as well as the evolution of research groups, particularly with the co-authorship formula. In this direction it would be convenient to distinguish this kind of social particularity in the collaboration network analysis.

Reviewer:

Is the article adequately referenced? Can be improved

Author:

The references were modified and corrected in the next way:

It was removed this bibliographic reference:

 3.-González Carrero, O.D. Diagnóstico y contextualización del sector porcino en el mundo para la consecución de buenas prácticas del modelo logístico de la cadena de suministro porcina, tesis de ingenieria. Licenciatura, Universidad Católica de Colombia: Bogotá, Colombia, 2019.

It was included this bibliographic reference:

  1. OECD/FAO (2021), OECD-FAO Agricultural Outlook 2021-2030, OECD Publishing, Paris, https://doi.org/10.1787/19428846-en.

It was removed this bibliographic reference:

  1. Carreras, M.R. Más de 100 artículos científicos conectan las enfermedades zoonóticas con la explotación animal. www.elsaltodiario.com 2021.

It was included this bibliographic reference:

  1. UPC (Centre for Animal Ethics). Resources on zoonotic diseases and their connection to animal exploitation. 2021. Available online: https://www.upf.edu/web/cae-center-for-animal-ethics/zoonotic-pandemics. (accessed on October 2021).

It was removed this bibliographic reference:

  1. Carreras, M.R. Virus, cerdos y humanos: nuestra adicción a comer animales y sus consecuencias. www.elsaltodiario.com 2020.

It was included this bibliographic reference:

  1. Sun H., Xiao Y., Liu J., Wang D., Li F., Wang Ch., Li Ch., Zhu J., Song J., Sun H., Jiang Z., Liu L., Wei K., Hou D., Pu J., Sun Y., Tong Q., Bi Y., Kin-Chow Ch., Liu S., Gao G.F., Liu J. Prevalent Eurasian avian-like H1N1 swine influenza virus with 2009 pandemic viral genes facilitating human infection., PNAS, 2020, 117, 29, https://doi.org/10.1073/pnas.1921186117.

It was removed this bibliographic reference:

  1. FIRA, F.I.R. con la A. Panorama Agroalimentario. Carne de cerdo 2020 2020.

It was included this bibliographic reference:

  1. Rabobank. Rabo Research Food & Agribusiness. Pork Quarterly Q4 2019: Producers Remain Cautious on Expansion as Risks Outweigh Rewards.2021. Available online: https://www.rabobankwholesalebankingna.com/pork-quarterly-q4-2019-producers-remain-cautious-on-expansion-as-risks-outweigh-rewards/ (accessed on 05 June 2021).

It was removed this bibliographic reference:

  1. Anderson, E.N. Everyone Eats. Understanding Food and Culture; 1st ed.; New York University Press: Nueva York, EUA, 2005; ISBN 0-8147-0495-6.

It was included this bibliographic reference:

  1. Falowo A.B. and Akimoladun O. F. Veterinary Drug Residues in Meat and Meat Products: Occurrence, Detection and Implications. Veterinary Medicine and Pharmaceuticals. 2018. DOI: 10.5772/intechopen.83616.

There were included the next bibliographic references:

  1. Bouvard V., Loomis D., Guyton K.Z., Grosse Y., Ghissassi F., Benbrahim-Tallaa L., Guha N.,  Mattock H., Straif K. Carcinogenicity of consumption of red and processed meat. Lancet Oncol. 2015, 16, 16. 1599-600. doi: 10.1016/S1470-2045(15)00444-1.
  2. Briggs H. Coronavirus: WHO developing guidance on wet markets. BBC Science, 2020. Available online: https://www.bbc.com/news/science-environment-52369878
  3. BBC News. Coronavirus: More work needed to rule out China lab leak theory says WHO. BBC China, 2020. Available online: https://www.bbc.com/news/world-asia-china-56581246
  4. Newman M. E. J. The structure of scientific collaboration networks. PNAS, 2001, 98, 2, 404-409. https://doi.org/10.1073/pnas.98.2.404.
  5. Popp J., Balogh P., Oláh J., Kot S., Harangi R . M. and Lengyel P. Social Network Analysis of Scientific Articles Published by Food Policy. Sustainability, 2018, 10, 577. doi:10.3390/su10030577.
  6. Barabási, A.-L.; Jeong, H.; Néda, Z.; Ravasz, E.; Schubert, A.; Vicsek, T. Evolution of the social network of scientific collaborations. Phys. A Stat. Mech. Appl. 2002, 311, 590–614. https://doi.org/10.1016/S0378-4371(02)00736-7.
  7. Börner, K.; Dall’Asta, L.; Ke, W.; Vespignani, A. Studying the emerging global brain: Analyzing and visualizing the impact of co-authorship teams. Complexity 2005, 10, 57–67. https://doi.org/10.1002/cplx.20078
  8. Chang, C.-L.; McAleer, M. Bibliometric rankings of journals based on the Thomson Reuters citations database. J. Rev. Glob. Econ. 2015, 4, 11, 120–125. DOI:10.6000/1929-7092.2015.04.11
  9. Gotian R. Networking for introverted scientists. Nature, 2019. DOI: 10.1038/d41586-019-01296-2.
  10. Joubert S. The Importance of Networking in Science. 2018 [blog]. Available online: https://www.northeastern.edu/graduate/blog/biotechnology-networking-tips/
  11. APA. Why it's important for you to present your data at scientific conferences. Psychological Science Agenda. November 2007. Available online: https://www.apa.org/science/about/psa/2007/11/student-council-1.
  12. Fistetti F. Comunidad. Léxico de la política. Buenos Aires, Nueva Visión. 2005.
  13. Rebollar-Rebollar, A.; Gómez-Tenorio, G.; Rebollar-Rebollar, S.; Hernández-Martínez, J.; González-Razo, F.J. Dinámica regional de la producción porcina en México, 1994-2012. Agrociencia 2015, 49,4, 455–473. Available online: https://agrociencia-colpos.mx/index.php/agrociencia/article/view/1158